# Synthesis and characterization of selenium nanoparticles stabilized with oxyethylated alkylphenol (neonol) for potential modification of fabric materials

Zafar Rekhman[1], Andrey Blinov[1], Alexey Gvozdenko[1], Alexey Golik[1], Andrey Nagdalian[1], Anastasia Blinova[1], Alexander Serov[1], Maxim Pirogov[1], Alina Askerova[1], Ekaterina Nazaretova[1], Mohammad Ali Shariati[2], Afnan A. Al Zahrani[3], Ammar AL-Farga[4], Saleh M. Al-maaqar[5]*

1 North Caucasus Federal University, Stavropol, Russia, 2 Scientific Department, Semey Branch of the Kazakh Research Institute of Processing and Food Industry, Almaty, Kazakhstan, 3 Department of Biology, Faculty of Science & Literature – Baljurshi, Al-Baha University, Al Bahah, Saudi Arabia, 4 Department of Biochemistry, College of Science, University of Jeddah, Jeddah, Saudia Arabia, 5 Department of Biology, Faculty of Education, Albaydha University, Al-Baydha, Yemen

* al-maaqar@baydaauniv.net

**Data Availability Statement:** All relevant data are within the manuscript.

## Abstract

This work demonstrates the first time synthesis of selenium nanoparticles (Se NPs) stabilized with neonol. The synthesis method was optimized using a multifactorial experiment with three input parameters. The most stable sample had a radius of 15 nm and a ζ-potential of -36.76 mV. It was found that the optimal parameters for the synthesis of Se NPs stabilized with neonol are the following concentration values: 0.12 mol/L selenic acid, 0.095 mol/L neonol and 0.95 mol/L ascorbic acid. Quantum chemical modeling of Se-neonol molecular complex formation showed that interaction of Se with neonol occurs through a hydroxyl group. Difference in the total energy of the neonol molecule and Se-neonol molecular complex is more than 2399 kcal/mol, which indicates that formation of chemical bond between Se and neonol is energetically advantageous. It was found that all samples exhibit stability over the entire pH range from 1.81 to 11.98, and the particle size is in the range of 25–30 nm. The analysis of the study of the influence of the ionic force showed that cations do not significantly affect the Se NPs radius, but anions have a significant effect, increasing the average hydrodynamic radius up to 2750 nm. For modification with Se NPs, silk, gauze, wool, cotton and cardboard samples were used. Elemental mapping of the samples showed an ambiguous distribution of Se NPs over the surface of fabric material. Assessment of potential antibacterial activity of modified fabric materials revealed inhibition zones of *Micrococcus luteus* growth from 12 to 16 mm for silk, gauze, wool and cotton. Notably, the most intense inhibition of *Micrococcus luteus* was observed in wool treated be Se NPs stabilized with neonol. Cardboard did not express *Micrococcus luteus* growth inhibition action because of weak interaction of cellulose filaments with Se NPs and neonol and possible microbial digestion of cellulose and xylan.

**Funding:** Funding was provided by the Russian Science Foundation (23-16-00120) to AB.

## 1. Introduction

Natural textile materials are a favorable media for the development and growth of bacteria, viruses and fungi [1, 2]. Therefore, modification of textile materials in order of formation of antimicrobial properties is critically important [3, 4]. Recent studies show that nanoparticles of metals and their oxides (silver, copper, selenium etc.) have broader antimicrobial properties compared to conventional preparations, and are used for processing textile materials in the form of ready-made dispersions [5–8]. However, considering practical application, modification of textile materials with nanoparticles does not ensure strong fixation and high resistance to physico-chemical influences [9–11].

Currently, there is growing interest in the synthesis and characterization of selenium nanoparticles (Se NPs), since this chemical element has unique photoelectric, semiconductor, catalytic and biological properties [12–15]. Obviously, the long-range forces between Se NPs and macromolecules (stabilizer), as well as between the resulting nanostructures, play a key role in the formation of functional nanocomplexes [16, 17]. The properties of nanostructures, not least, depend on the phenomena on the interphase surfaces of nanoparticle-stabilizer, nanoparticle-solvent and stabilizer-solvent [18]. There is also an inverse relationship: the morphology of polymer nanosystems determines the nature of the processes of formation of nanostructures [19]. These factors largely determine the unique properties of Se-based nanocomposites and, ultimately, their application. For instance, modification of bioactive packaging films with Se NPs can inhibit weight loss, oxidative browning, and the emergence of black mold on the packaged cloves [20]. It is known about the use of Se NPs—containing nanocomposite to remove mercury ions from wastewater [21].

In the absence of stabilizers, Se NPs in aqueous solutions are aggregatively unstable [22, 23]. One of the most promising ways to obtain stable Se NPs is restoring their ionic forms in polymer solutions. At the same time, during pseudomatric synthesis, mutual "recognition" of macromolecules and emerging Se NPs occurs, which ensures control of their sizes by varying the structure and molecular weight of polymers [24, 25]. Sodium selenite, selenic acid, or sodium selenosulfate can be used as selenium-containing precursors [26, 27]. Non-toxic reagents such as ascorbic acid, sugars, and amino acids are usually used as reducing agents, but the resulting particles are not stable [28, 29]. In order to control their shape and size, the obtained particles are stabilized by adding substances such as chitosan, polyvinyl alcohol, and surfactants [30–32]. Notably, Se NPs can be obtained by green synthesis methods [33]. For instance, Puri et al. [34] produced Se NPs using *Terminalia arjuna* bark extract and included them in a gel for biomedical applications. The authors revealed that Se NPs exhibit antioxidant, antibacterial and anticancer activity.

For use in the pulp and paper and textile industries, one of the most promising Se NPs stabilizers may be neonol (oxyethylated alkylphenol), which belongs to nonionic surfactants [35]. Neonol is used in the textile industry as the active base of technical (commercial) detergents, and is the starting material for the synthesis of some types of active base of textile reclamation agents [36, 37]. Neonol has high detergent properties and the ability to retain impurities in solution even without additional additives, is characterized by chemical resistance in hard water and good compatibility with auxiliary components of the detergent mixture [38, 39].

As far as we know, there are no precedents in the scientific literature for the synthesis of Se NPs stabilized with neonol. Therefore, there is a scientific and practical interest in the synthesis and characterization of the Se-neonol nanocomplex. Thus, the purpose of this study was synthesis and characterization of Se NPs stabilized with neonol, as well as assessment of the potential antibacterial properties of textile materials modified with prepared Se NPs.

## 2. Materials and methods

### 2.1. Synthesis of Se NPs

Synthesis of Se NPs was carried out by chemical reduction in an aqueous medium in the presence of stabilizer. Neonol (Vitareactive, Dzerzhinsk, Russia)–a nonionic surfactant from the class of oxyethylated alkylphenols was used as a stabilizer. Selenious acid (INTRERHIM, St. Petersburg, Russia) was used as a selenium-containing precursor, while ascorbic acid (Lenreactive, St. Petersburg, Russia) was used as a reducing agent.

For the synthesis, solutions with a different ratio of neonol to selenious acid were prepared. To do this, 0.68 g neonol were dissolved in 100 mL of 0.036 M selenious acid solution, depending on the set ratio. Then, 0.088M ascorbic acid solution was prepared by dissolving 773.8 mg of ascorbic acid in 50 mL of distilled water. Consequently, an ascorbic acid solution was added drop by drop to a solution of selenious acid and a neonol with intensive stirring and the resulting solution was mixed for 5–10 minutes.

### 2.2. Optimization of Se NPs synthesis

To optimize the experimental parameters of Se NPs synthesis, a multifactorial experiment was performed with three input parameters and three levels of variation. The output parameters were: the average hydrodynamic radius of the particles (R) and the $\zeta$-potential.

The study of the average hydrodynamic radius of Se NPs was carried out using the dynamic light scattering (DLS) method with Photocor-Complex device (Antek-97 LLC, Moscow, Russia) and DynaLS software. For measurement, samples of Se NPs were diluted 4 times with distilled water. The $\zeta$-potential of Se NPs was studied by acoustic and electroacoustic spectroscopy using DT-1202 device (Dispersion Technology Inc., USA).

Using preliminary experiments, the levels of variation of variables (concentrations of selenious acid, neonol and ascorbic acid) were estimated. The resulting data are shown in Table 1.

Based on data of Table 1, the experiment matrix was compiled and presented in Table 2.

The mathematical processing of the experimental results and neural network creation were carried out in the Neural Statistica Network application software package (Tulsa, US).

### 2.2. Assessment of Se NPs stability

To study the effect of pH on the stability of positive and negative sols of Se NPs, solutions with different pH in a ratio of 1:1 were added to the prepared samples. To prepare sodium acetate buffer solutions with different pH a 0.04 solution of phosphoric, acetic and boric acids was prepared. For this, 5.49 mL $H_3PO_4$, 4.58 mL $CH_3COOH$ and 4.95 g $H_3BO_3$ were mixed in volumetric flask and the volume of solution was adjusted to 2 L with distilled water. 700 mL of 0.2 M NaOH solution was prepared separately. Then, to obtain a sodium acetate buffer solution with the required pH, X mL of 0.2 M NaOH solution was added to 100 mL of the acids solution. The volumes of NaOH used for the corresponding pH values are shown in Table 3.

To study the effect of the ionic strength of the solution on the stability of Se NPs, five series were prepared: solutions of sodium chloride (NaCl), iron chloride ($FeCl_3$), barium chloride

**Table 1. Levels of variable variation.**

| Reagent | Sign of parameter | Levels of variable variation | | |
|---|---|---|---|---|
| (selenious acid), mol/L | a | 0.004 | 0.120 | 0.236 |
| (neonol), mol/L | b | 0.006 | 0.180 | 0.355 |
| (ascorbic acid), mol/L | c | 0.032 | 1.076 | 2.12 |

**Table 2. The experiment matrix.**

| Sample | a (mol/L) | b (mol/L) | c (mol/L) |
|---|---|---|---|
| 1 | 0.004 | 0.006 | 0.032 |
| 2 | 0.004 | 0.180 | 1.076 |
| 3 | 0.004 | 1.076 | 2.12 |
| 4 | 0.120 | 0.006 | 1.076 |
| 5 | 0.120 | 0.180 | 2.12 |
| 6 | 0.120 | 1.076 | 0.032 |
| 7 | 0.236 | 0.006 | 2.12 |
| 8 | 0.236 | 0.180 | 0.033 |
| 9 | 0.236 | 1.076 | 1.076 |

**Table 3. Volumes of NaOH and corresponding pH values.**

| NaOH volume, mL | pH |
|---|---|
| 0 | 1.81 |
| 10 | 2.21 |
| 20 | 3.29 |
| 30 | 4.56 |
| 40 | 5.72 |
| 50 | 6.8 |
| 60 | 7.96 |
| 70 | 9.15 |
| 80 | 10.38 |
| 90 | 11.58 |
| 100 | 11.98 |

($BaCl_2$), sodium sulfate ($Na_2SO_4$), potassium phosphate ($K_3PO_4$). The concentrations of each solution varied of 0.1 M, 0.25 M, 0.5 M, 0.75 M and 1 M. To assess the stability, 1 mL of Se NPs sol was added to 9 mL of each solution. Table 4 shows the method of solutions preparation.

## 2.3. Quantum chemical modelling

Quantum chemical modeling of the interaction of Se NPs with neonol was carried out using QChem software with IQmol molecular editor at the following construction parameters: calculation—Energy, method—B3LYP, basis– 6-31G*, convergence– 5, force field—Chemical [40]. The calculation was carried out on the equipment of the data processing center (Schneider Electric) of North Caucasus Federal University.

The models obtained were tested for the total energy of the molecular complex (E), the difference in the total energy of the neonol molecule and Se-neonol molecular complex (ΔE), the energy of the highest occupied molecular orbital ($E_{HOMO}$), the energy of the lowest unoccupied molecular orbital ($E_{LUMO}$) and the chemical hardness of the system (η) calculated by formula (1):

$$\eta = \frac{E_{LUMO} - E_{HOMO}}{2}.$$

(1)

**Table 4. Preparation of solutions for study of the effect of ionic strength on Se NPs stability.**

| Salts | Concentration | | | | |
|---|---|---|---|---|---|
| | 0.1 | 0.25 | 0.5 | 0.75 | 1 |
| | Weight, g | | | | |
| NaCl | 0.029 | 0.073 | 0.146 | 0.219 | 0.29 |
| $Na_2SO_4$ | 0.071 | 0.178 | 0.355 | 0.53 | 0.71 |
| $K_3PO_4$ | 0.082 | 0.205 | 0.41 | 0.615 | 0.82 |
| $FeCl_3$ | 0.08 | 0.203 | 0.406 | 0.609 | 0.812 |
| $BaCl_2$ | 0.103 | 0.257 | 0.514 | 0.77 | 1.03 |

## 2.4. Fourier-transform infrared spectroscopy

Fourier-transform infrared spectroscopy (FTIR) was performed to study the functional groups in the obtained samples. IR-spectra were recorded by FSM-1201 spectrometer (Infraspek, St. Petersburg, Russia). The measurement range was 500–4000 $cm^{-1}$.

## 2.5. Modification of fabric materials with Se NPs

Cotton, medical gauze, cardboard, silk, and natural wool were used as fabric materials. Materials were cut into $1 \times 1$ $cm^2$. The procedure for applying Se NPs was performed as follows: each material was placed in a reaction medium, held for 5 minutes and dried at room temperature for 24 h.

The microstructure and elemental mapping of modified fabric materials were studied using a MIRA-LMH scanning electron microscope (SEM) with the AZtecEnergy Standard/X-max 20 (standard) elemental composition determination system (Tescan, Brno-Kohoutovice, Czech Republic).

## 2.6. Assessment of potential bactericidal activity of modified fabric materials

Assessment of potential bactericidal activity of modified fabric materials was carried out on Gram-positive bacteria *Micrococcus luteus*. To prepare the microbial suspension, the *Micrococcus luteus* culture was washed off the agar with sterile tap water. The resulting suspension was filtered through a sterile cotton-gauze filter and diluted with sterile tap water to a concentration of $\sim 1.5 \times 10^8$ in 1 mL which is equal to 0.5 units according to the McFarland standard, determined using a densitometer [41].

The prepared nutrient medium was poured into sterile Petri dishes. To obtain a uniform bacterial lawn on the surface of the agar, 1 mL of the suspension of the test culture was poured into the Petri dishes.

On the surface of the seeded agar, at a distance of 2 cm from the edge of the Petri dishes and at the same distance from each other, disks of the studied samples were laid out with tweezers.

## 2.7. Statistically data processing

The experiments were carried out in threefold biological and fivefold analytical repetition. All parameters obtained were submitted to one-way analysis of variance (ANOVA) and Student's T-test ($p < 0.05$) through the statistical package STATISTICA for Windows (Statsoft, Tulsa,

USA). Microsoft Excel 2010 and Origin software were also used for histograms and graphs creation based on the results of the data processing.

## 3. Result and discussion

Photon correlation spectroscopy of Se NPs (Fig 1A) showed that samples 2 (4217 nm), 4 (1739 nm), 5 (6041 nm), 6 (6631 nm) had the largest average hydrodynamic radius. The smallest Se NPs were observed in samples 1 (16 nm), 7 (15 nm), 8 (20 nm) and 9 (29 nm). In turn, analysis of $\zeta$-potential of samples (Fig 1B) revealed that sample 7 with the lowest value of the lowest value of $\zeta$-potential (-36.76 mV) was the most stable [42]. Consequently, this sample was selected for study of the effect of ionic strength and pH on Se NPs stability.

As a result of mathematical data processing, a three-dimensional ternary dependence was obtained (Fig 1C). Analysis of the ternary surface showed that concentrations of the precursor, stabilizer and reducing agent have a significant effect on the average hydrodynamic radius of Se NPs, which size varied from 20 to 3000 nm. According to the results obtained, the optimal parameters for the synthesis of Se NPs stabilized with neonol are the following concentration values: 0.12 mol/L selenic acid, 0.095 mol/L neonol, 0.95 mol/L ascorbic acid.

Results of quantum chemical modelling are presented in Fig 2. It was found that the total energy of the neonol molecule is -2199.528 kcal/mol, and the energy of Se-neonol molecular complex is -4599.173 kcal/mol. Such a high difference in the total energy (2399 kcal/mol) indicates the energy benefit of the formation of a chemical bond between Se and neonol [43]. Chemical stability of the molecular complex is evidenced by the value of the chemical hardness of the system [44]. Quantum chemical calculations revealed that $\eta$ (Se-Neonol) = 0.078, which indicates high chemical stability. According to the modelling, interaction occurs through a hydroxyl group.

At the next stage, Se NPs stabilized with neonol were examined by FTIR spectroscopy. The obtained IR spectra are shown in Fig 3.

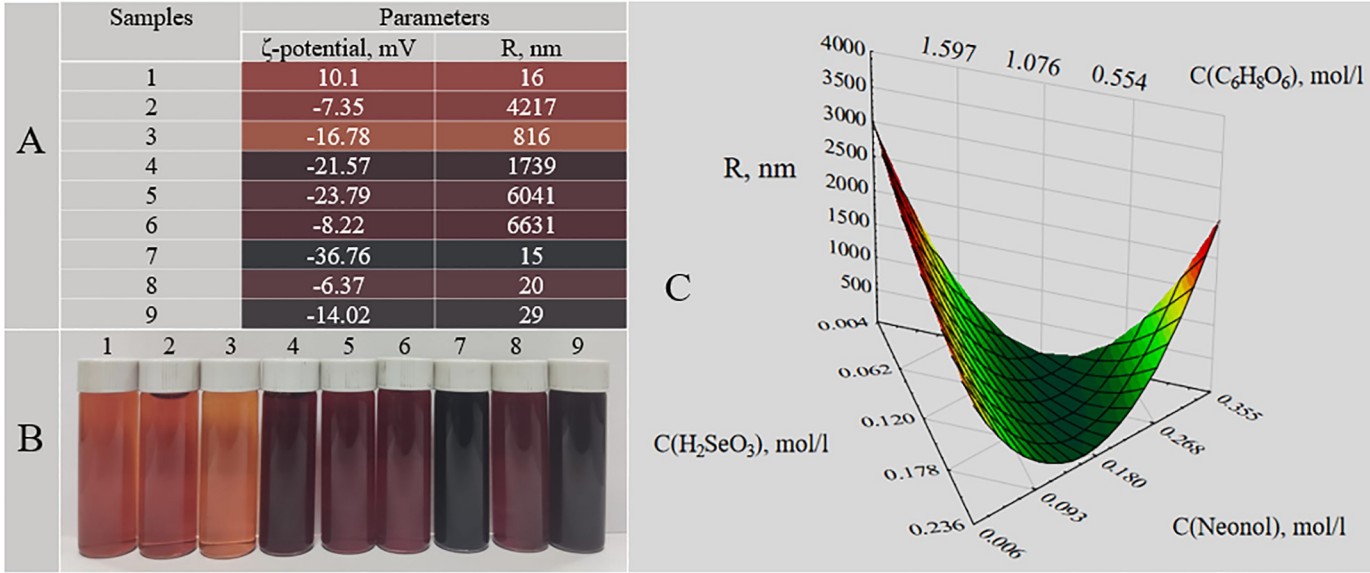

**Fig 1.** Optimization of Se NPs synthesis: values of average hydrodynamic radius and $\zeta$-potential of 9 samples obtained (A), visualization of 9 samples obtained (B), the ternary surface of dependences of average hydrodynamic radius on concentrations of stabilizer and reducing agent (C).

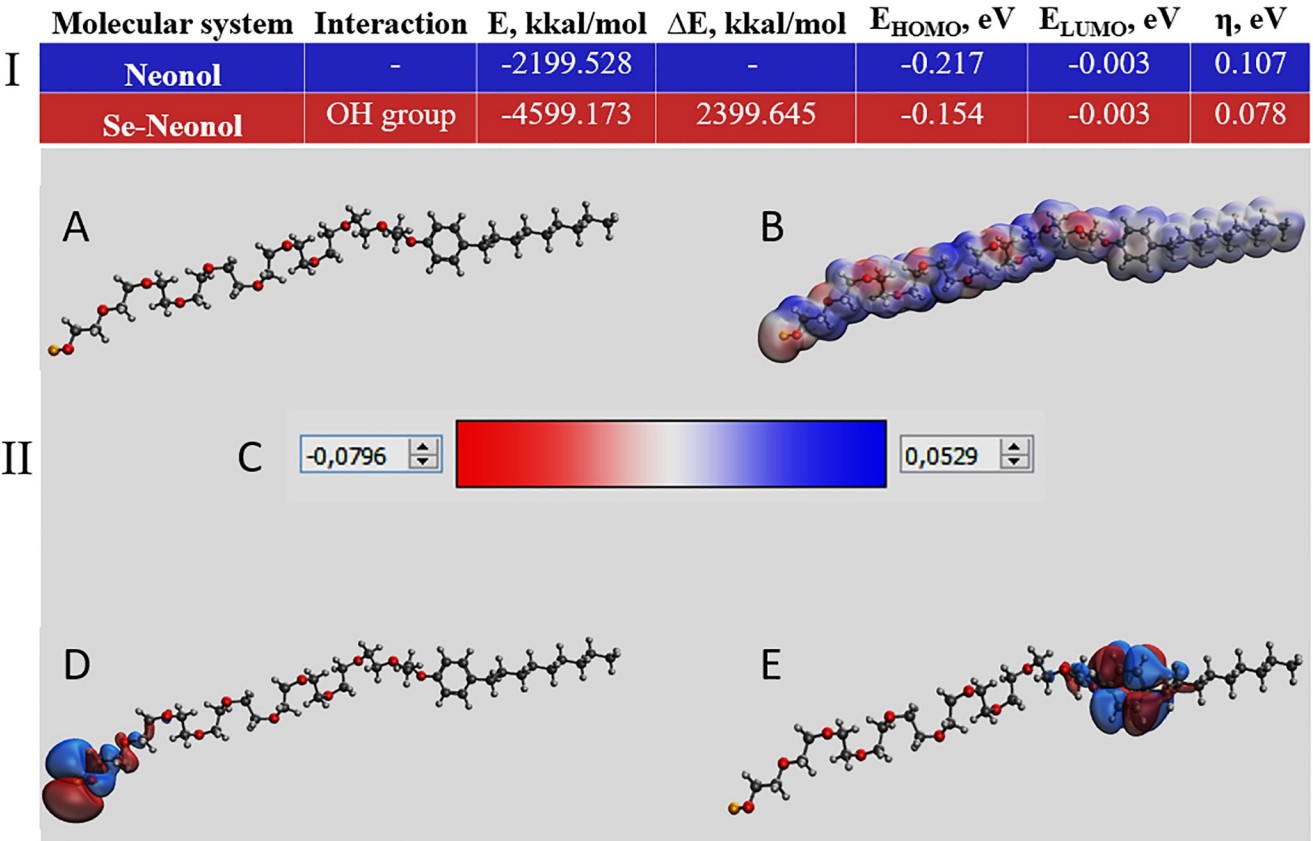

| | Molecular system | Interaction | E, kkal/mol | $\Delta E$, kkal/mol | $E_{HOMO}$, eV | $E_{LUMO}$, eV | $\eta$, eV |
|---|---|---|---|---|---|---|---|
| I | Neonol | - | -2199.528 | - | -0.217 | -0.003 | 0.107 |
| | Se-Neonol | OH group | -4599.173 | 2399.645 | -0.154 | -0.003 | 0.078 |

**Fig 2. Quantum chemical modelling of Se-neonol molecular complex.** I—Quantum chemical calculations. II—Quantum chemical models: molecular complex model (A), electron density distribution (B), electron density distribution gradient (C), the highest occupied molecular orbital HOMO (D), the lowest unoccupied molecular orbital LUMO (E).

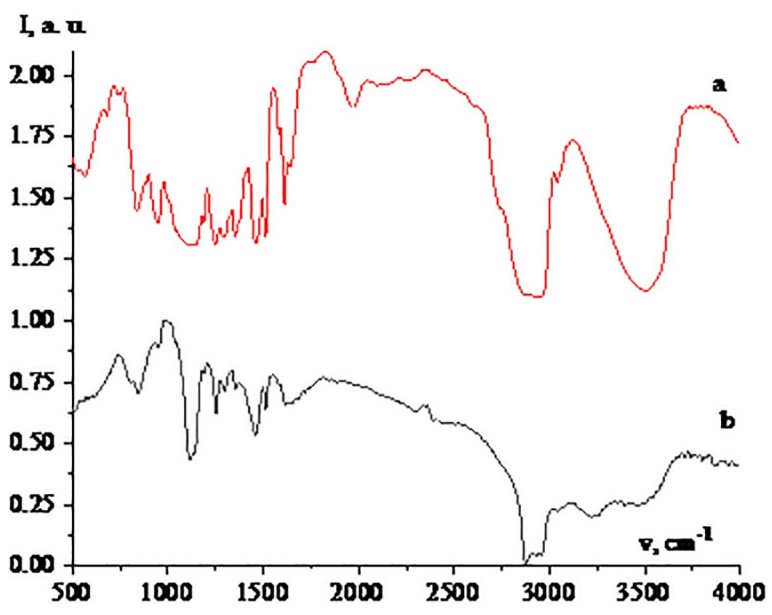

**Fig 3. IR-spectra of neonol (a) and Se NPs stabilized with neonol (b).**

Analysis of the IR spectrum of neonol showed that in the range from 500 to 1800 cm$^{-1}$ there are deformation vibrations of -CH (562, 683 cm$^{-1}$), -CH2 (741, 827 cm$^{-1}$), -CH3 (942, 1012 cm$^{-1}$) groups, vibrations of C-O (1197–1353 cm$^{-1}$), C-C (1469.1526 cm$^{-1}$) bonds, as well as C-C and C = C bonds in the benzene ring (1584–1642 cm$^{-1}$). In the range from 2600 to 4000 cm$^{-1}$, the presence of bands characteristic of valence vibrations of CH$_2$ groups (2860–2976 cm$^{-1}$) and O-H bond (3518 cm$^{-1}$) is observed [45–49].

Analysis of the IR spectrum of Se NPs stabilized with neonol showed that in the range from 500 to 1800 cm$^{-1}$ there are low-intensity oscillations of Se-Se bond (562 cm$^{-1}$), deformation vibrations of -CH$_2$ (790.845 cm$^{-1}$) and -CH$_3$ (954 cm$^{-1}$) groups, as well as vibrations of C-O (1110–1347 cm$^{-1}$), C-C (1457, 1520 cm$^{-1}$) bonds and C-C, C = C bonds in the benzene ring (1613–1653 cm$^{-1}$). In the range from 2600 to 4000 cm$^{-1}$, the presence of bands characteristic of valence vibrations of -CH$_2$ (2840–3024 cm$^{-1}$) and -CH (3180–3300 cm$^{-1}$) groups and O-H bond (3407–3593 cm$^{-1}$) is observed [45–49].

Thus, it can be concluded that interaction of Se NPs and neonol occurs through a hydroxyl group, which is consistent with the results of computer quantum chemical modeling.

A study of the effect of pH on the stability of Se NPs stabilized with neonol (Fig 4A and 4B) showed that samples exhibit aggregative stability over the entire pH range. Thus, the average hydrodynamic radius of Se NPs varied from 20 to 25 nm. The results obtained correspond to findings of previous works of synthesis and characterization of Se NPs stabilized with cocamidopropyl betaine [24] and Catamine AB [26].

At the next stage, the effect of various ions on Se NPs sol stability was investigated. The obtained photographs showing the coagulating effect of salts and the dependence of the average hydrodynamic radius of Se NPs on concentration of cations and anions are shown in Fig 4C–4E. The analysis of the data obtained showed that negative ions have a significant effect on

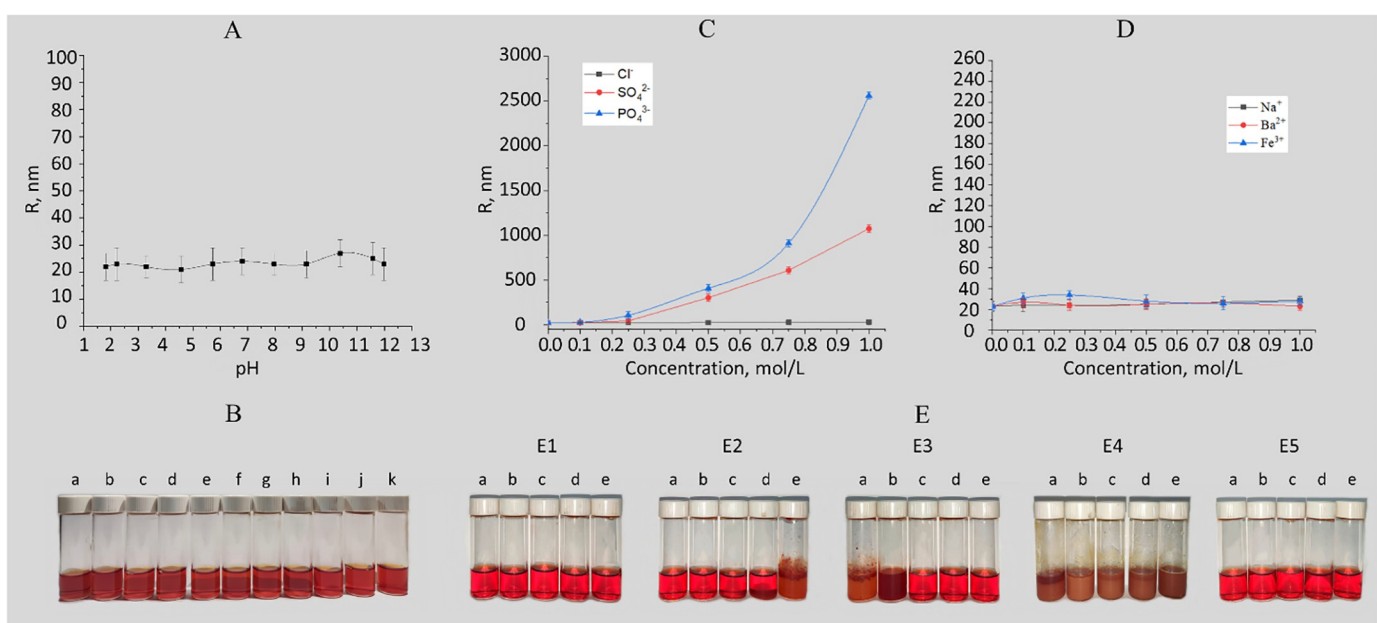

**Fig 4.** Stability of Se NPs stabilized with neonol: A—Dependence of the average hydrodynamic radius on pH. B—Visualization of samples obtained with set pH: 1.81 (a), 2.21 (b), 3.29 (c), 4.56 (d), 5.72 (e), 6.8 (f), 7.96 (g), 9.15 (h), 10.38 (i), 11.58 (j), 11.98 (k). C—Dependence of the average hydrodynamic radius on concentration of anions. D—Dependence of the average hydrodynamic radius on concentration of cations. E—Visualization of Se-Neonol samples with NaCl (E1), Na$_2$SO$_4$ (E2), Na$_3$PO$_4$ (E3), FeCl$_3$ (E4), BaCl$_2$ (E5) at concentrations of 0.1M (a), 0.25M (b), 0.5M (c), 0.75M (d), 1.0 M (e).

stability of Se NPs stabilized with neonol. It is important to note, that the greater the charge of an ion opposite to the charge of Se NPs, the more pronounced the effect it has on particle aggregation, which is consistent with the Schulze-Hardy rule [50]. Thus, $Cl^-$ ions practically did not affect the radius of Se NPs. For $SO_4^{2-}$ and $PO_4^{3-}$ ions, an increase in ionic strength led to an increase in Se NPs radius. Notably, increase in $SO_4^{2-}$ concentration to 1 mol/L led to increase of Se NPs radius to 1000 nm. Moreover, the analysis of the dependence for $PO_4^{3-}$ ion showed that an increase in concentration leads to a more significant increase in the radius of colloidal particles up to 2750 nm. It is worth noting that for $SO_4^{2-}$ and $PO_4^{3-}$ ions, turbidity of solutions and precipitation were observed throughout the concentration range.

From another side, it was found that cations do not have a significant effect on the coagulation of Se NPs stabilized with neonol. As can be seen in Fig 4D, when NaCl, $FeCl_2$ and $FeCl_3$ salts are added to Se NPs sol, the average hydrodynamic radii were not changed and had a value of 25–30 nm. However, at the same time, when the salts were added, Se NPs samples became cloudy, and after a while the particles coagulated Fig 4E.

Thus, the experiment showed that Se NPs stabilized with neonol are relatively stable at changing conditions. Thus, there was a practical interest to test synthesized Se NPs at modification of fabric materials. For this, samples of silk, gauze, wool, cotton and cardboard were treated with Se-neonol nanocomplex and studied using SEM with energy dispersive X-ray spectroscopy (EDS). SEM micrographs and elemental mapping of modified surfaces of fabric materials are presented in Fig 5.

According to Fig 5, all materials consist of filaments with a thickness of 10 to 20 μm. Distribution of Se NPs in samples had a different degree of uniformity. The most uniform distribution of Se NPs was observed on the surface of silk filaments. Uneven Se NPs distribution was revealed on the surface of cardboard, natural wool and gauze filaments, which led to formation of areas with increased concentration of Se NPs. Cotton filaments contained separate aggregates of Se NPs.

Most likely, discovered trends are related to the physico-chemical properties and chemical nature of studied materials. Thus, silk is of animal origin and consists of the proteins fibroin (hydrophobic component) and sericin (hydrophilic component), which is located on the surface of silk filaments [51]. Sericin has 18 amino acids in its sequence, among which glycine, alanine, tyrosine, serine, and lysine can be distinguished [52]. Amino acids located on the surface of the silk filaments can enter into chemical interactions with the hydrophilic part of the neonol [53, 54]. The possible scheme of interaction of Se NPs stabilized with neonol with surface of silk filaments is visualized and presented in Fig 6.

Adsorption of Se NPs stabilized with neonol on the surface of natural wool filaments is hindered by their hydrophobicity. However, EDS results provide strong evidence for the formation of Se NPs on the surface of the natural wool filaments, which is in the line with findings of Elmaaty et al. in studies of wool modification and dying with Se NPs stabilized by ascorbic acid and polyvinylpyrrolidone [55, 56].

In turn, gauze, cardboard and cotton are fabric materials consisting of cellulose [57]. As known, cellulose does not have positively charged groups in its composition, which can chemically interact with negatively charged groups in the hydrophilic part of neonol [58]. Notably, better distribution of Se NPs on surface of synthetic materials was achieved at use of ascorbic acid [59, 60].

The next step of the experiment was assessment of potential bactericidal activity of modified fabric materials on Gram-positive bacteria *Micrococcus luteus* culture (Fig 7). The choice of *Micrococcus luteus* is justified by its spread distribution on skin and fabric materials. Recent studies show that that 80% of all micrococci of skin and fabric materials are representatives of the *luteus* species [61, 62]. At the same time, *Micrococcus luteus* are potentially harmful at weak immunity [63].

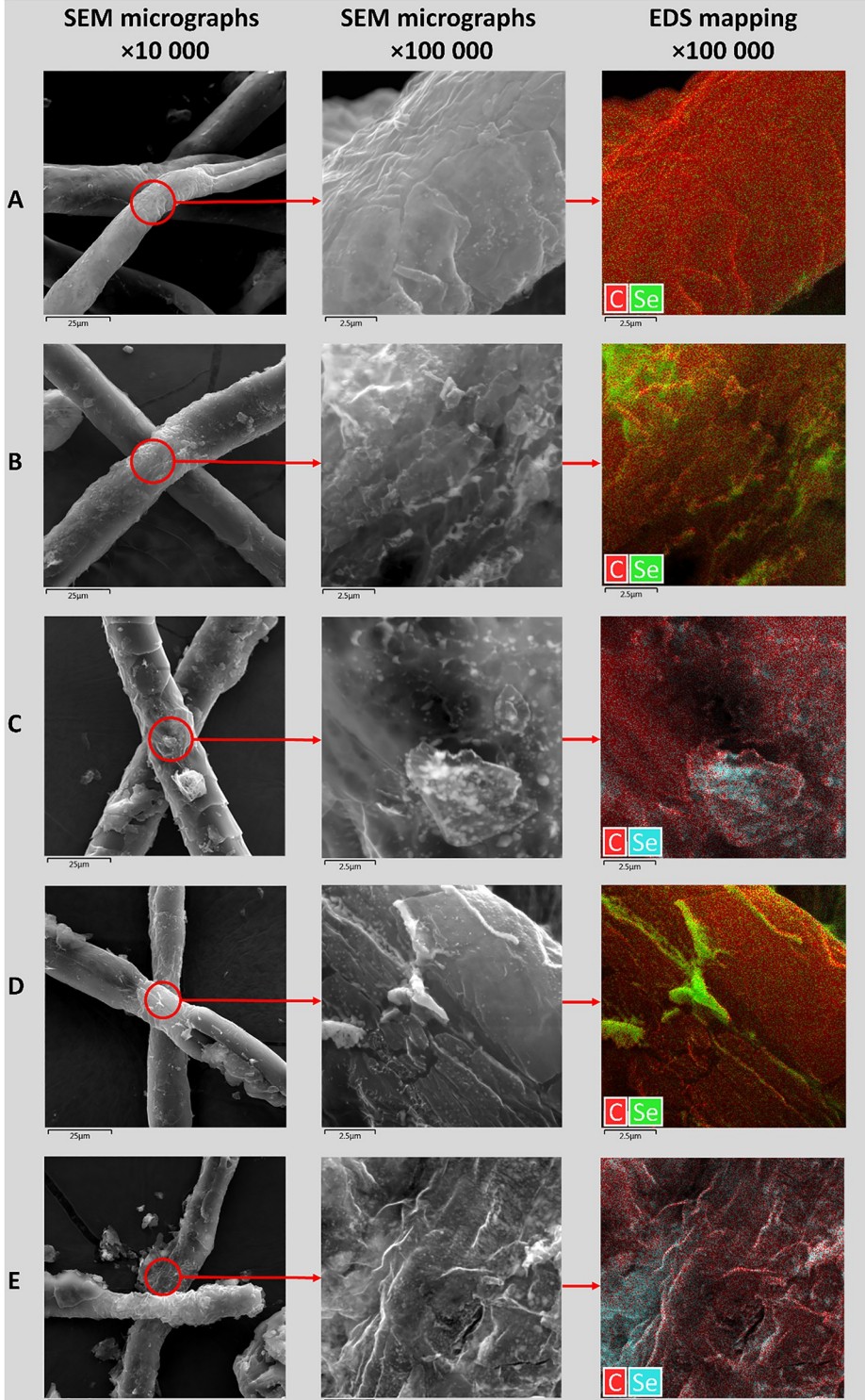

**Fig 5. Characterization of microstructure of silk (A), gauze (B), wool (C), cotton (D) and cardboard (E) modified with Se NPs stabilized with neonol, using scanning electron microscopy (SEM micrographs) with energy dispersive X-ray spectroscopy (EDS mapping).**

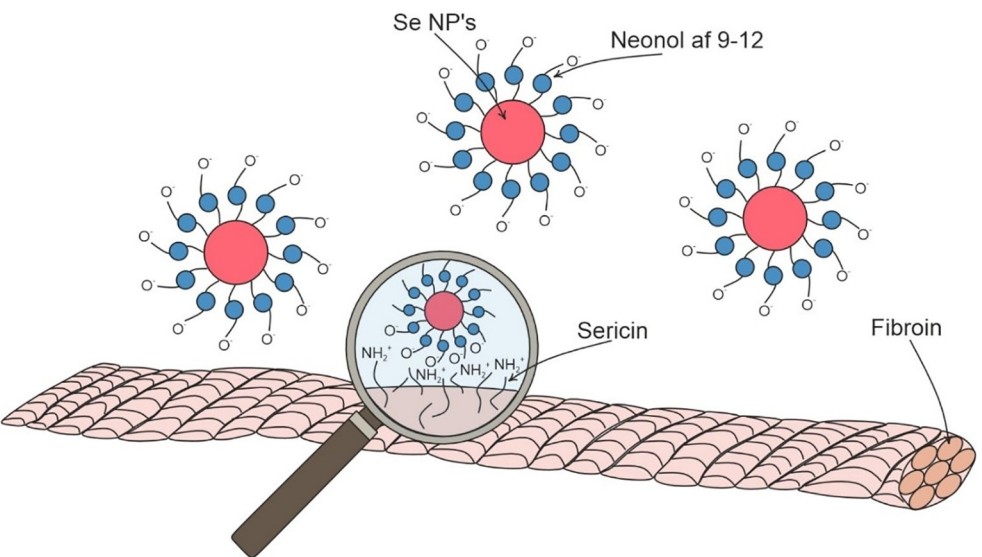

**Fig 6. Scheme of binding of Se NPs stabilized with neonol to the surface of silk filaments.**

The analysis of the results obtained revealed different inhibition zones (IZ) of *Micrococcus luteus* depending on the type of fabric material treated with Se NPs. According to Fig 7, modified natural wool has the highest antibacterial potential (IZ = 16 mm) in comparison with other samples, which indicates the lowest resistance of *Micrococcus luteus* to this material. It is important to note, that modified cotton, gauze and silk showed IZ of 12, 13 and 15 mm, respectively, which also characterize potential antibacterial activity of produced materials. The results obtained are in the line with previous studies of antimicrobial activity of Se NPs stabilized with different surface chemistry and structure [64, 65]. Notably, the revealed antibacterial activity of Se NPs stabilized with neonol against *Micrococcus luteus* is slightly lower than the results obtained for Se NPs stabilized with cetyltrimethylammonium chloride [24]. However, it is worth noting that this study examined the antibacterial activity of tissue material modified with Se NPs, while previous study tested antibacterial activity of Se NPs solution.

Recent report of Mirza et al. [58] confirms antimicrobial activity of cotton with Se NPs against *S. aureus* (IZ = 32 mm), *E. coli* (IZ = 16 mm) and *K. pneumoniae* (IZ = 26 mm). Similar results were observed by Elmaaty et al. [66] who recorded IZ of 15 mm for *B. cereus*, 8 mm for *E. coli*, 18 mm for *P. Aeruginosa* and 23 mm for *S. typhi* testing polyster treated with 25 mM Se NPs. Similarly, Yip et al. [67] demonstrated that the spacer fabric treated with green synthesized Se NPs can inhibit more than 99% of *T. rubrum* and *S. aureus* growth at first 12 h of explication.

The main exception is modified cardboard, that did not express growth inhibition action to *Micrococcus luteus*. Apparently, this is due to weak interaction of cellulose filaments with Se NPs and neonol and, at the same time, possible microbial digestion of cellulose and xylan as a crucial factor for *Micrococcus luteus* growth [68, 69].

It is worth noting, that question of the mechanism of bacterial activity of Se NPs is currently open. According to literature data, Se NPs can induce apoptosis, can stimulate the production of reactive oxygen species, can cause damage to mitochondria, destruction of cell membranes, and even interruption of transmembrane electron transport [70–72]. Se NPs stimulate production of reactive oxygen species, which can be adsorbed on the surface of tissue materials and provide additional antibacterial activity [13].

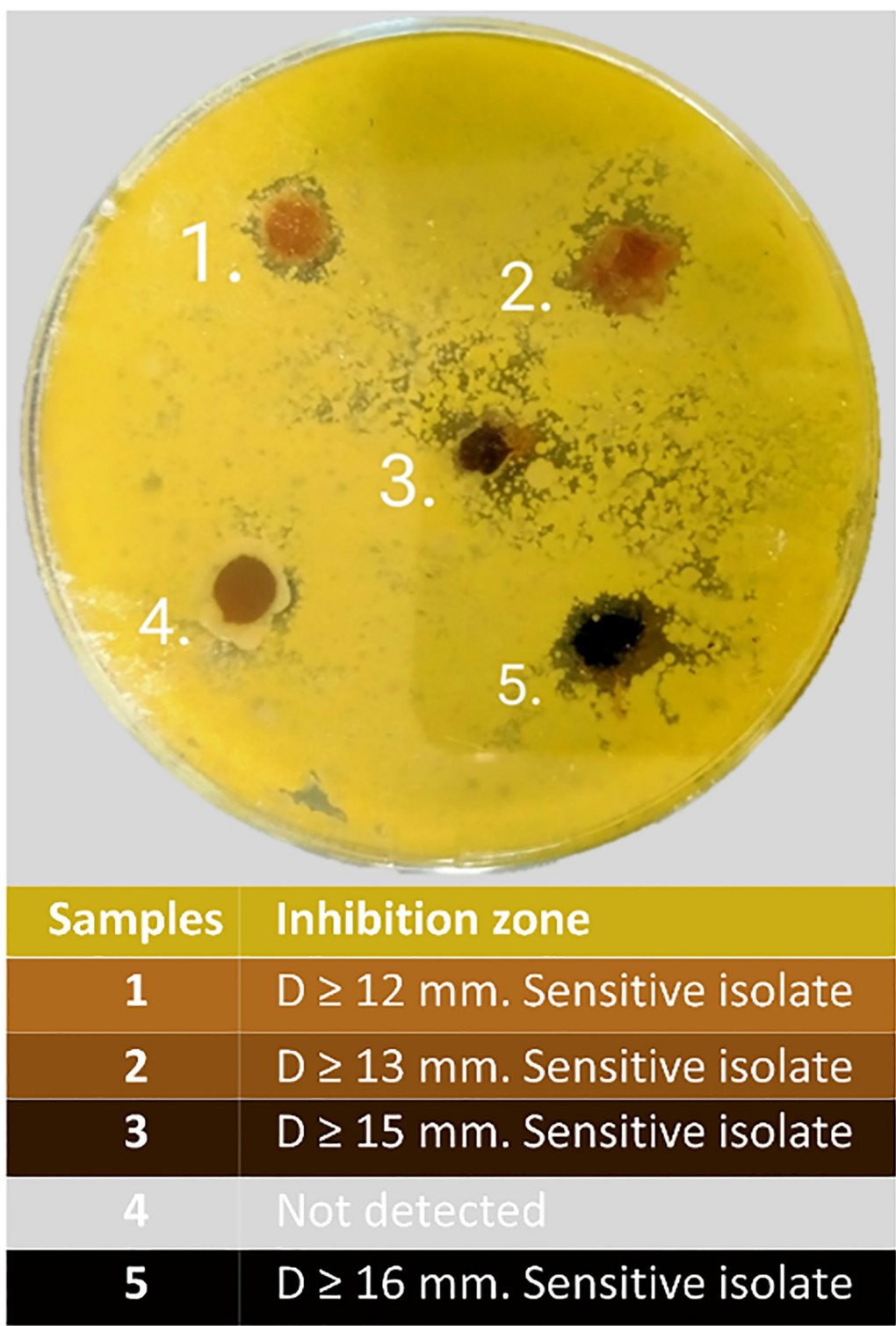

| Samples | Inhibition zone |
|---------|-----------------|
| 1 | D ≥ 12 mm. Sensitive isolate |
| 2 | D ≥ 13 mm. Sensitive isolate |
| 3 | D ≥ 15 mm. Sensitive isolate |
| 4 | Not detected |
| 5 | D ≥ 16 mm. Sensitive isolate |

**Fig 7. Assessment of potential bactericidal activity of modified fabric materials on *Micrococcus luteus* culture:** Cotton (1), gauze (2), silk (3), cardboard (4), natural wool (5).

It can be assumed that Se NPs penetrate into bacterial cells through the cell membrane and start exerting their bacteriostatic effect on it. This occurs due to the disruption of the cell wall and penetration of Se NPs into the cell [73]. Se NPs can break the phospholipid bilayer, interact with intracellular proteins and inactivate them or react with sulfhydryl and thiol groups of membrane proteins, eventually denaturing them [73, 74].

In this regard, it can be concluded that modification of fabric materials such as wool, gauze, cotton and silk by Se NPs stabilized with neonol can form new antibacterial properties in these materials. Such materials have a great potential for application in clothes, uniforms, medical means, antibacterial coatings and packaging. Meanwhile, the toxicity of fabric materials treated with Se NPs stabilized with neonol should be comprehensively studied before industrial application. Further, more in-depth and long-term studies should be done to understand the mechanisms of potential antibacterial and antifungal activity of Se NPs stabilized with neonol. Finally, it is critically important for future works to understand potential migration of Se NPs and neonol from modified fabric materials to skin or products depending on way of application.

## 4. Conclusions

This work demonstrates the first time synthesis of Se NPs stabilized with neonol. The synthesis method was optimized using a multifactorial experiment with three input parameters. The most stable sample had a radius of 15 nm and a $\zeta$-potential of -36.76 mV. It was found that the optimal parameters for the synthesis of Se NPs stabilized with neonol are the following concentration values: 0.12 mol/L selenic acid, 0.095 mol/L neonol and 0.95 mol/L ascorbic acid. Quantum chemical modeling of Se-neonol molecular complex formation showed that interaction of Se with neonol occurs through a hydroxyl group. Difference in the total energy of the neonol molecule and Se-neonol molecular complex is more than 2399 kcal/mol, which indicates that formation of chemical bond between Se and neonol is energetically advantageous. Notably, all samples exhibited stability over the entire pH range from 1.81 to 11.98 with constant size of 25–30 nm. Similarly, various concentrations of cations ($Na^+$, $Ba^{2+}$, $Fe^{3+}$) had no effect on the average hydrodynamic radius of Se NPs. In turn, anions had a significant effect on Se NPs: 1 mol/L $SO_4^{2-}$ ions led to increase in particle size up to 1000 nm, and 1 mol/L $PO_4^{3-}$ ions–up to 2750 nm. All changes were accompanied by the appearance of turbidity and coagulation of particles.

For modification with Se NPs, silk, gauze, wool, cotton and cardboard samples were used. Elemental mapping of the samples showed that all materials consist of filaments with a thickness of 10 to 20 μm. Se NPs were distributed unevenly. The best distribution of Se NPs was observed on the silk surface. Other samples had uneven distribution of Se NPs or contained either aggregates of Se NPs in separate regions.

Assessment of potential antibacterial activity of modified fabric materials revealed inhibition zones of *Micrococcus luteus* growth from 12 to 16 mm for silk, gauze, wool and cotton. Notably, the most intense inhibition of *Micrococcus luteus* was observed in wool treated be Se NPs stabilized with neonol. Cardboard did not express *Micrococcus luteus* growth inhibition action, which is, apparently, due to weak interaction of cellulose filaments with Se NPs and neonol and, at the same time, possible microbial digestion of cellulose and xylan as a crucial factor for *Micrococcus luteus* growth.

Thus, modification of fabric materials such as wool, gauze, cotton and silk by Se NPs stabilized with neonol can form new antibacterial properties in these materials. Such materials have a great potential for application in clothes, uniforms, medical means, antibacterial coatings and packaging. However, this is a first study of modification of fabric materials by Se NPs

stabilized with neonol and further studies are important to understand potential toxicity of synthesized Se NPs, its potential antibacterial and antifungal activity against wide range of microorganisms and possible migration of Se NPs from the surface of treated fabric materials to skin or product depending on was of application.

## Author Contributions

**Conceptualization:** Zafar Rekhman, Andrey Blinov, Andrey Nagdalian, Alexander Serov, Mohammad Ali Shariati.

**Data curation:** Alina Askerova.

**Investigation:** Alexey Gvozdenko, Maxim Pirogov, Alina Askerova, Ekaterina Nazaretova.

**Methodology:** Zafar Rekhman, Andrey Blinov, Ekaterina Nazaretova, Afnan A. Al Zahrani, Saleh M. Al-maaqar.

**Software:** Zafar Rekhman, Alexey Golik.

**Supervision:** Ammar AL-Farga.

**Validation:** Anastasia Blinova.

**Visualization:** Alexey Golik, Maxim Pirogov.

**Writing – original draft:** Zafar Rekhman, Andrey Blinov, Alexey Gvozdenko.

**Writing – review & editing:** Zafar Rekhman, Alexey Golik, Andrey Nagdalian, Anastasia Blinova, Alexander Serov, Mohammad Ali Shariati, Afnan A. Al Zahrani, Ammar AL-Farga, Saleh M. Al-maaqar.

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
