## [Decision Letter · Decision Letter 0]

19 Jun 2024

PONE-D-24-17851Synthesis and characterization of selenium nanoparticles stabilized with oxyethylated alkylphenol (neonol) for potential modification of fabric materialsPLOS ONE

Dear Dr. Al-maaqar,

Thank you for submitting your manuscript to PLOS ONE. After careful consideration, we feel that it has merit but does not fully meet PLOS ONE’s publication criteria as it currently stands. Therefore, we invite you to submit a revised version of the manuscript that addresses the points raised during the review process. Please submit your revised manuscript by Aug 03 2024 11:59PM. If you will need more time than this to complete your revisions, please reply to this message or contact the journal office at plosone@plos.org. Please include the following items when submitting your revised manuscript:A rebuttal letter that responds to each point raised by the academic editor and reviewer(s). You should upload this letter as a separate file labeled 'Response to Reviewers'.A marked-up copy of your manuscript that highlights changes made to the original version. You should upload this as a separate file labeled 'Revised Manuscript with Track Changes'.An unmarked version of your revised paper without tracked changes. You should upload this as a separate file labeled 'Manuscript'.

We look forward to receiving your revised manuscript.

Kind regards,

Nayan Ranjan Singha, Ph.D.

Academic Editor

PLOS ONE

2. We note that Figure 5 in your submission contain copyrighted images. All PLOS content is published under the Creative Commons Attribution License (CC BY 4.0), which means that the manuscript, images, and Supporting Information files will be freely available online, and any third party is permitted to access, download, copy, distribute, and use these materials in any way, even commercially, with proper attribution. For more information, see our copyright guidelines: http://journals.plos.org/plosone/s/licenses-and-copyright.

1. You may seek permission from the original copyright holder of Figure 5 to publish the content specifically under the CC BY 4.0 license.

Reviewers' comments:

Reviewer's Responses to Questions

**Comments to the Author**

1. Is the manuscript technically sound, and do the data support the conclusions?

Reviewer #1: Yes

Reviewer #2: Yes

Reviewer #3: Yes

2. Has the statistical analysis been performed appropriately and rigorously? 

Reviewer #1: Yes

Reviewer #2: Yes

Reviewer #3: Yes

3. Have the authors made all data underlying the findings in their manuscript fully available?

Reviewer #1: Yes

Reviewer #2: Yes

Reviewer #3: Yes

4. Is the manuscript presented in an intelligible fashion and written in standard English?

Reviewer #1: Yes

Reviewer #2: Yes

Reviewer #3: Yes

5. Review Comments to the Author

Reviewer #1: Te paper reports the results of the synthesis and characterization of selenium nanoparticles stabilized with neonol, as well as antimicrobial properties of various fabric materials treated with these nanoparticles. The manuscript sounds scientific, the results are obtained using adequate methods and statistically treated. It was interesting to read the paper. I have two comments listed below.

1. A correlation analysis would be useful between the antimicrobial activity of SeNP-modified fabric materials and the distribution of SeNPs on their surfaces estimated by SEM-EDS.

2. How the authors would explain the antimicrobial activity of SeNP-modified fabric materials if nanoparticles are strongly immobilized on their surfaces, thus hindering the interactions with bacterial cells? Some positive and negative controls should be performed with SeNPs and untreated fabric materials.

Reviewer #2: Based on the manuscript titled: Synthesis and characterization of selenium nanoparticles stabilized with oxyethylated alkylphenol (neonol) for potential modification of fabric materials, the following should be addressed

1. The introduction can be enriched by citing the followings articles: DOI: 10.1016/j.ijbiomac.2023.128073; DOI: 10.3389/fchem.2023.1273360

2. The FTIR of neonol and neonol stabilized selenium nanoparticles should be provided.

3. The name of the buffer solution prepared using Table 3 should be provided.

Reviewer #3: In this article, the author has demonstrated the first-time synthesis of selenium nanoparticles (Se NPs) stabilized with neonol and their antibacterial activities. It is interesting work indicating some valuable characteristics in studied nanoparticles. However, some points should be considered before its possible publication in the Journal as below:

1. in Experimental section, the authors should indicate exact amount of used reagent. For example, 0.68 g - 5.24 g neonol is not precise.

2. Used “mL” unit instead of “cm3 and dm3” in whole manuscript.

3. In Table 1. Use “reagent” instead of “parameter”

4. In Table 1 place concentration unit in front of reagent; E.g. reagent concentration (mol/L)

5. In table 1, delete (mol/L) unite from “Levels of variable variation (mol/L)”

6. In order to study leaching and/or changes in Se NPs size, after catalytic performance, either TEM, FTIR, XRD or EDX test is required. See literature reports for more information as follow:

https://doi.org/10.1002/aoc.7286

https://doi.org/10.1002/aoc.7191

7. The term "chemical rigidity" is not a standard concept in chemistry. It seems to be a term that is not widely used or recognized in the field so Use “chemical hardness” instead of “chemical rigidity”.

8. List starting materials used in the study with their sources.

9. Chemical hardness plays an important role in chemical interactions. It is a concept derived from quantum chemistry that quantifies the resistance of a chemical species to electron transfer. The authors should further discuss this interesting parameter with update references in this regard:

https://doi.org/10.1007/s00214-023-03076-8

10. As part of the work, the authors should explain the density functional theory (DFT) as proposed method and the reason why the use B3LYP/6-31G* as specific computational protocol. it's worth noting that the choice of computational method and basis set can depend on the specific system under investigation and the desired level of accuracy. Different combinations may be more appropriate for different types of molecules or properties of interest. you could refer to latest relevant literature reports:

https://doi.org/10.1007/s11270-023-06447-w

11. The introduction section is rather plain, the author should impart the importance and application of nanoparticles and nanocomposite including Se nanoparticles in areas such as water treatment, antibacterial, environmental remediation and organic pollutants degradation with the help from the suggested literature reports:

https://doi.org/10.1038/s41598-019-43368-3

https://doi.org/10.1016/j.jenvman.2020.111263

12. How does the composition of ascorbic acid reduce Se ions to their corresponding nanoparticles? The literature contains several similar useful works as below:

https://doi.org/10.1038/s41598-019-43368-3

https://doi.org/10.1002/star.201500347

https://doi.org/10.1038/s41598-021-85832-z

8. From the industrial and commercialization point of view, Can Se NPs be scaled up for large-scale production?

9. Green synthesis as novel technique often utilizes natural, renewable resources and employ eco-friendly reaction conditions as seen in current work. Hence, in separate paragraph authors should discuss further common biological routes for the biosynthesis of nanoparticles including Se NPs using natural resources such as Plant extracts, Microorganism among others. The following literature references are helpful:

https://doi.org/10.1038/s41598-019-43368-3

10. Grammatical and typos errors are observed throughout the manuscript.

11. in figure 2, page 7, Asian the peaks around 2 eV in EDX analysis.

12. To determine the functional groups in Se-neonol structure FTIR test is required.

https://doi.org/10.1016/j.ceramint.2018.02.091

15. mechanism of photodegradation steps have poorly explained and incomplete. To obtain in-depth perception for reaction Mechanistic sketch (step by step) is required. See aforementioned references.

16. For comprehension, the authors should compare the antibacterial efficiency of Se NPs against pollutants in dark, UV, and visible conditions.

https://doi.org/10.1515/gps-2019-0040

https://doi.org/10.1007/s11270-023-06356-y

19. What is the mechanism of bactericidal function?

20. identify the bacteria type tested in this study

22. To have comprehension view, tabulate similar literature report including current results.

6. PLOS authors have the option to publish the peer review history of their article (what does this mean?). If published, this will include your full peer review and any attached files.

Reviewer #1: No

Reviewer #2: No

Reviewer #3: No

---

## [Author Response · Author response to Decision Letter 0]

9 Oct 2024

Reviewer 1

We are grateful to the Reviewer 1 for his/her positive evaluation and for the time devoted to review our manuscript. All comments were useful and pleased us with the high level of understanding of the topic. We have addressed all recommendations as requested. All changes in the manuscript are marked by green. Please see the point-by-point answers below.

1. A correlation analysis would be useful between the antimicrobial activity of SeNP-modified fabric materials and the distribution of SeNPs on their surfaces estimated by SEM-EDS.

Response: Thank you for your suggestion! Correlation analysis between the antimicrobial activity of Se NP-modified fabric materials and the distribution of Se NPs on their surfaces, as estimated by SEM-EDS, could indeed provide valuable insights and strengthen the work. However, it is unfortunately not possible to carry out this analysis with the current results. We are unable to quantify the content of particles on the surface of the material using the techniques available to us. While there was an option to analyze the activity based on the selenium content from the EDS spectrum, this would have required consideration during the initial experimental design stage. Therefore, we appreciate your suggestion and will consider incorporating it in future studies.

2. How the authors would explain the antimicrobial activity of SeNP-modified fabric materials if nanoparticles are strongly immobilized on their surfaces, thus hindering the interactions with bacterial cells? Some positive and negative controls should be performed with SeNPs and untreated fabric materials.

Response: Thank you for your suggestion! Correlation analysis between the antimicrobial activity of Se NP-modified fabric materials and the distribution of Se NPs on their surfaces, as estimated by SEM-EDS, could indeed provide valuable insights and strengthen the work. However, it is unfortunately not possible to carry out this analysis with the current results. We are unable to quantify the content of particles on the surface of the material using the techniques available to us. While there was an option to analyze the activity based on the selenium content from the EDS spectrum, this would have required consideration during the initial experimental design stage. Therefore, we appreciate your suggestion and will consider incorporating it in future studies.

Reviewer 2

We are grateful to the Reviewer 2 for his/her positive evaluation and for the time devoted to review our manuscript. All comments were useful and pleased us with the high level of understanding of the topic. We have addressed all recommendations as requested. All changes in the manuscript are marked by green. Please see the point-by-point answers below.

Based on the manuscript titled: Synthesis and characterization of selenium nanoparticles stabilized with oxyethylated alkylphenol (neonol) for potential modification of fabric materials, the following should be addressed

1. The introduction can be enriched by citing the followings articles: 

DOI: 10.1016/j.ijbiomac.2023.128073; DOI: 10.3389/fchem.2023.1273360 

Response: Thank you for suggestion of such useful works. We studied them and used to strengthen Introduction section. 

2. The FTIR of neonol and neonol stabilized selenium nanoparticles should be provided. 

Response: Thank you for suggestion. We carried out FTIR spectroscopy. The results were added to the manuscript.

3. The name of the buffer solution prepared using Table 3 should be provided.

Response: Corrected.

Reviewer 3

We are grateful to the Reviewer 3 for his/her positive evaluation and for the time devoted to review our manuscript. All comments were useful and pleased us with the high level of understanding of the topic. We have addressed all recommendations as requested. All changes in the manuscript are marked by green. Please see the point-by-point answers below.

In this article, the author has demonstrated the first-time synthesis of selenium nanoparticles (Se NPs) stabilized with neonol and their antibacterial activities. It is interesting work indicating some valuable characteristics in studied nanoparticles. However, some points should be considered before its possible publication in the Journal as below:

1. in Experimental section, the authors should indicate exact amount of used reagent. For example, 0.68 g - 5.24 g neonol is not precise. 

Response: Revised

2. Used "mL" unit instead of "cm3 and dm3" in whole manuscript. 

Response: Revised

3. In Table 1. Use "reagent" instead of "parameter" 

Response: Revised

4. In Table 1 place concentration unit in front of reagent; E.g. reagent concentration (mol/L) 

Response: Revised

5. In table 1, delete (mol/L) unite from "Levels of variable variation (mol/L)"

Response: Revised

6. In order to study leaching and/or changes in Se NPs size, after catalytic performance, either TEM, FTIR, XRD or EDX test is required. 

See literature reports for more information as follow:

https://eur02.safelinks.protection.outlook.com/?url=https%3A%2F%2Fdoi.org%2F10.1002%2Faoc.7286&data=05%7C02%7Camalfarga%40uj.edu.sa%7Cfedc3392bbbe461e6a5a08dc915021f2%7Cb453d91b6ac14b61b8b85e65e422233f%7C0%7C0%7C638545020560912240%7CUnknown%7CTWFpbGZsb3d8eyJWIjoiMC4wLjAwMDAiLCJQIjoiV2luMzIiLCJBTiI6Ik1haWwiLCJXVCI6Mn0%3D%7C0%7C%7C%7C&sdata=ee%2FunViA%2BmaJ1OE5j65n5rAsWMS20gjAwHu4fJc5KtQ%3D&reserved=0

https://eur02.safelinks.protection.outlook.com/?url=https%3A%2F%2Fdoi.org%2F10.1002%2Faoc.7191&data=05%7C02%7Camalfarga%40uj.edu.sa%7Cfedc3392bbbe461e6a5a08dc915021f2%7Cb453d91b6ac14b61b8b85e65e422233f%7C0%7C0%7C638545020560916951%7CUnknown%7CTWFpbGZsb3d8eyJWIjoiMC4wLjAwMDAiLCJQIjoiV2luMzIiLCJBTiI6Ik1haWwiLCJXVCI6Mn0%3D%7C0%7C%7C%7C&sdata=2kC5X7jMZtM3eoCYhxta4qBmnop%2BeRgg3ywba9RLIzE%3D&reserved=0

Response: Thank you for your recommendation. We have additionally carried out FTIR analysis and incorporated the results into the manuscript. Notably, TEM analysis is planned for the next experiment within the same project. We also conducted XRD analysis, but the resulting diffractograms indicated that the samples are too amorphous for this type of characterization to be included in the manuscript. This finding aligns with the results obtained in our previous works, where XRD results were also not usable (https://doi.org/10.1038/s41598-022-25884-x, https://doi.org/10.3390/nano13243128, https://doi.org/10.1038/s41598-023-33581-6, https://keypublishing.org/jhed/wp-content/uploads/2021/04/26.-JHED-Volume-34-FFP-Full-paper-Andrey-Blinov.pdf). In addition, we would like to thank the Reviewer for suggesting relevant works, which we have already cited in the manuscript.

7. The term "chemical rigidity" is not a standard concept in chemistry.

It seems to be a term that is not widely used or recognized in the field so Use "chemical hardness" instead of "chemical rigidity". 

Response: Revised

8. List starting materials used in the study with their sources. 

Response: The starting materials used in the synthesis are given in subsection 2.1

9. Chemical hardness plays an important role in chemical interactions.

It is a concept derived from quantum chemistry that quantifies the resistance of a chemical species to electron transfer. The authors should further discuss this interesting parameter with update references

in this regard:

https://eur02.safelinks.protection.outlook.com/?url=https%3A%2F%2Fdoi.org%2F10.1007%2Fs00214-023-03076-8&data=05%7C02%7Camalfarga%40uj.edu.sa%7Cfedc3392bbbe461e6a5a08dc915021f2%7Cb453d91b6ac14b61b8b85e65e422233f%7C0%7C0%7C638545020560921597%7CUnknown%7CTWFpbGZsb3d8eyJWIjoiMC4wLjAwMDAiLCJQIjoiV2luMzIiLCJBTiI6Ik1haWwiLCJXVCI6Mn0%3D%7C0%7C%7C%7C&sdata=t6hGRG0m%2BZOnfwNb%2Fyr27Gjq75BpHK2N6lvNHGzr3F8%3D&reserved=0

Response: Thank you for suggestion. This study was considered in the revised manuscript. 

10. As part of the work, the authors should explain the density functional theory (DFT) as proposed method and the reason why the use B3LYP/6-31G* as specific computational protocol. it's worth noting that the choice of computational method and basis set can depend on the specific system under investigation and the desired level of accuracy. Different combinations may be more appropriate for different types of molecules or properties of interest. you could refer to latest relevant literature reports:

https://eur02.safelinks.protection.outlook.com/?url=https%3A%2F%2Fdoi.org%2F10.1007%2Fs11270-023-06447-w&data=05%7C02%7Camalfarga%40uj.edu.sa%7Cfedc3392bbbe461e6a5a08dc915021f2%7Cb453d91b6ac14b61b8b85e65e422233f%7C0%7C0%7C638545020560926272%7CUnknown%7CTWFpbGZsb3d8eyJWIjoiMC4wLjAwMDAiLCJQIjoiV2luMzIiLCJBTiI6Ik1haWwiLCJXVCI6Mn0%3D%7C0%7C%7C%7C&sdata=fqd9qGJLdXY3akycIFubM1wZ31xeSv5KfnpWXbc7wxY%3D&reserved=0

Response: The use of the B3LYP method and the 6-31G* basis is associated with the need to calculate the electron density distribution.

11. The introduction section is rather plain, the author should impart the importance and application of nanoparticles and nanocomposite including Se nanoparticles in areas such as water treatment, antibacterial, environmental remediation and organic pollutants degradation with the help from the suggested literature reports:

https://eur02.safelinks.protection.outlook.com/?url=https%3A%2F%2Fdoi.org%2F10.1038%2Fs41598-019-43368-3&data=05%7C02%7Camalfarga%40uj.edu.sa%7Cfedc3392bbbe461e6a5a08dc915021f2%7Cb453d91b6ac14b61b8b85e65e422233f%7C0%7C0%7C638545020560930928%7CUnknown%7CTWFpbGZsb3d8eyJWIjoiMC4wLjAwMDAiLCJQIjoiV2luMzIiLCJBTiI6Ik1haWwiLCJXVCI6Mn0%3D%7C0%7C%7C%7C&sdata=crdAuDJbM6P1k3GoXkFrYjvTQ6GLBTRQ3HBvJ%2FV0dAQ%3D&reserved=0

https://eur02.safelinks.protection.outlook.com/?url=https%3A%2F%2Fdoi.org%2F10.1016%2Fj.jenvman.2020.111263&data=05%7C02%7Camalfarga%40uj.edu.sa%7Cfedc3392bbbe461e6a5a08dc915021f2%7Cb453d91b6ac14b61b8b85e65e422233f%7C0%7C0%7C638545020560935535%7CUnknown%7CTWFpbGZsb3d8eyJWIjoiMC4wLjAwMDAiLCJQIjoiV2luMzIiLCJBTiI6Ik1haWwiLCJXVCI6Mn0%3D%7C0%7C%7C%7C&sdata=5X9dT46%2BPA9ZqDHuCvq0KkjZGnHDEurS3Sbhl2RpM70%3D&reserved=0

Response: thank you for valuable notes. The Introduction section has been expanded to incorporate the suggested references.

12. How does the composition of ascorbic acid reduce Se ions to their corresponding nanoparticles? The literature contains several similar useful works as below:

https://eur02.safelinks.protection.outlook.com/?url=https%3A%2F%2Fdoi.org%2F10.1038%2Fs41598-019-43368-3&data=05%7C02%7Camalfarga%40uj.edu.sa%7Cfedc3392bbbe461e6a5a08dc915021f2%7Cb453d91b6ac14b61b8b85e65e422233f%7C0%7C0%7C638545020560940188%7CUnknown%7CTWFpbGZsb3d8eyJWIjoiMC4wLjAwMDAiLCJQIjoiV2luMzIiLCJBTiI6Ik1haWwiLCJXVCI6Mn0%3D%7C0%7C%7C%7C&sdata=EbIQ9cJq1rvvpI0IklUrh5aQCrgUnMc2s7SO%2FeuZ3cQ%3D&reserved=0

https://eur02.safelinks.protection.outlook.com/?url=https%3A%2F%2Fdoi.org%2F10.1002%2Fstar.201500347&data=05%7C02%7Camalfarga%40uj.edu.sa%7Cfedc3392bbbe461e6a5a08dc915021f2%7Cb453d91b6ac14b61b8b85e65e422233f%7C0%7C0%7C638545020560944926%7CUnknown%7CTWFpbGZsb3d8eyJWIjoiMC4wLjAwMDAiLCJQIjoiV2luMzIiLCJBTiI6Ik1haWwiLCJXVCI6Mn0%3D%7C0%7C%7C%7C&sdata=P%2FpIEbh2qS9qSmwJlxX%2B3ttCxlDrIhDOALaztBQungk%3D&reserved=0

https://eur02.safelinks.protection.outlook.com/?url=https%3A%2F%2Fdoi.org%2F10.1038%2Fs41598-021-85832-z&data=05%7C02%7Camalfarga%40uj.edu.sa%7Cfedc3392bbbe461e6a5a08dc915021f2%7Cb453d91b6ac14b61b8b85e65e422233f%7C0%7C0%7C638545020560949541%7CUnknown%7CTWFpbGZsb3d8eyJWIjoiMC4wLjAwMDAiLCJQIjoiV2luMzIiLCJBTiI6Ik1haWwiLCJXVCI6Mn0%3D%7C0%7C%7C%7C&sdata=MKsr%2Fgb2PF6gDt7ZvoTOYZ444WKHtg5%2B3kmmJOLpl6I%3D&reserved=0

Response: thank you. When ascorbic acid is added to a solution containing selenic acid, a redox reaction occurs, as a result of which selenic acid is reduced to elemental selenium.

13. From the industrial and commercialization point of view, Can Se NPs be scaled up for large-scale production?

Response: Certainly. This process can be implemented and will be evaluated in future studies.

14. Green synthesis as novel technique often utilizes natural, renewable resources and employ eco-friendly reaction conditions as seen in current work. Hence, in separate paragraph authors should discuss further common biological routes for the biosynthesis of nanoparticles including Se NPs using natural resources such as Plant extracts, Microorganism among others. The following literature references are helpful:

https://eur02.safelinks.protection.outlook.com/?url=https%3A%2F%2Fdoi.org%2F10.1038%2Fs41598-019-43368-3&data=05%7C02%7Camalfarga%40uj.edu.sa%7Cfedc3392bbbe461e6a5a08dc915021f2%7Cb453d91b6ac14b61b8b85e65e422233f%7C0%7C0%7C638545020560954172%7CUnknown%7CTWFpbGZsb3d8eyJWIjoiMC4wLjAwMDAiLCJQIjoiV2luMzIiLCJBTiI6Ik1haWwiLCJXVCI6Mn0%3D%7C0%7C%7C%7C&sdata=2V4bWpvOGCvgQNA6dKNg2FqQcJbj%2BfwhHPBAewYDX3Y%3D&reserved=0

Response: Thank you, added

15. Grammatical and typos errors are observed throughout the manuscript. 

Response: Thank you, revised.

16. in figure 2, page 7, Asian the peaks around 2 eV in EDX analysis. 

Response: In Figure 2, the EHOMO, ELUMO, and chemical hardness are expressed in eV. These values are derived from quantum chemical calculations performed using IQmol software.

17. To determine the functional groups in Se-neonol structure FTIR test is required.

https://eur02.safelinks.protection.outlook.com/?url=https%3A%2F%2Fdoi.org%2F10.1016%2Fj.ceramint.2018.02.091&data=05%7C02%7Camalfarga%40uj.edu.sa%7Cfedc3392bbbe461e6a5a08dc915021f2%7Cb453d91b6ac14b61b8b85e65e422233f%7C0%7C0%7C638545020560958846%7CUnknown%7CTWFpbGZsb3d8eyJWIjoiMC4wLjAwMDAiLCJQIjoiV2luMzIiLCJBTiI6Ik1haWwiLCJXVCI6Mn0%3D%7C0%7C%7C%7C&sdata=tJU13JUfH%2FUs8gLsIq6i3C637gb%2F3UVleZotAt87I%2FU%3D&reserved=0

Response: FTIR analysis has been incorporated into the work, and the discussion section has been expanded to include the suggested references.

18. mechanism of photodegradation steps have poorly explained and incomplete. To obtain in-depth perception for reaction Mechanistic sketch (step by step) is required. See aforementioned references.

Response: Thank you for the comment. However, photodegradation assessment was not considered in this work and is not planned for the next studies. 

19. For comprehension, the authors should compare the antibacterial efficiency of Se NPs against pollutants in dark, UV, and visible conditions.

https://eur02.safelinks.protection.outlook.com/?url=https%3A%2F%2Fdoi.org%2F10.1515%2Fgps-2019-0040&data=05%7C02%7Camalfarga%40uj.edu.sa%7Cfedc3392bbbe461e6a5a08dc915021f2%7Cb453d91b6ac14b61b8b85e65e422233f%7C0%7C0%7C638545020561101897%7CUnknown%7CTWFpbGZsb3d8eyJWIjoiMC4wLjAwMDAiLCJQIjoiV2luMzIiLCJBTiI6Ik1haWwiLCJXVCI6Mn0%3D%7C0%7C%7C%7C&sdata=O6Hi5yUy2JEu9%2FOAtO1VGzANk2C127TUeRJUMJMxBcQ%3D&reserved=0

https://eur02.safelinks.protection.outlook.com/?url=https%3A%2F%2Fdoi.org%2F10.1007%2Fs11270-023-06356-y&data=05%7C02%7Camalfarga%40uj.edu.sa%7Cfedc3392bbbe461e6a5a08dc915021f2%7Cb453d91b6ac14b61b8b85e65e422233f%7C0%7C0%7C638545020561109394%7CUnknown%7CTWFpbGZsb3d8eyJWIjoiMC4wLjAwMDAiLCJQIjoiV2luMzIiLCJBTiI6Ik1haWwiLCJXVCI6Mn0%3D%7C0%7C%7C%7C&sdata=93NjVy1q1ossggqGUWTgiXD5pBuA21LttlWn9zQUivI%3D&reserved=0

Response: Thank you for your comment. However, photodegradation assessment was not included in this work and is not planned for future studies.

20. What is the mechanism of bactericidal function?

Response: Thank you for your comment. The mechanism underlying the bacterial activity of selenium nanoparticles remains an open question. Current literature indicates that selenium nanoparticles may induce apoptosis, stimulate the production of reactive oxygen species, damage mitochondria, disrupt cell membranes, and interfere with transmembrane electron transport. We addressed aspects of the antimicrobial mechanism of Se NPs in our previous work (https://doi.org/10.3390/nano13243128). Our future studies aim to provide a more comprehensive understanding of the antimicrobial mechanisms of Se NPs.

21. identify the bacteria type tested in this study

Response: We worked with Gram positive bacteria. Now it is mentioned in the manuscript.

22. To have comprehension view, tabulate similar literature report including current results

Response: Thank you for your suggestion. We

---

## [Decision Letter · Decision Letter 1]

26 Oct 2024

PONE-D-24-17851R1Synthesis and characterization of selenium nanoparticles stabilized with oxyethylated alkylphenol (neonol) for potential modification of fabric materialsPLOS ONE

Dear Dr. Al-maaqar,Thank you for submitting your manuscript to PLOS ONE. After careful consideration, we feel that it has merit but does not fully meet PLOS ONE’s publication criteria as it currently stands. Therefore, we invite you to submit a revised version of the manuscript that addresses the points raised during the review process.

We look forward to receiving your revised manuscript.

Kind regards,

Nayan Ranjan Singha, Ph.D.

Academic Editor

PLOS ONE

Journal Requirements:

Reviewers' comments:

Reviewer's Responses to Questions

**Comments to the Author**

1. If the authors have adequately addressed your comments raised in a previous round of review and you feel that this manuscript is now acceptable for publication, you may indicate that here to bypass the “Comments to the Author” section, enter your conflict of interest statement in the “Confidential to Editor” section, and submit your "Accept" recommendation.

Reviewer #1: (No Response)

Reviewer #2: All comments have been addressed

Reviewer #3: All comments have been addressed

2. Is the manuscript technically sound, and do the data support the conclusions?

Reviewer #1: Partly

Reviewer #2: Yes

Reviewer #3: Yes

3. Has the statistical analysis been performed appropriately and rigorously? 

Reviewer #1: Yes

Reviewer #2: Yes

Reviewer #3: Yes

4. Have the authors made all data underlying the findings in their manuscript fully available?

Reviewer #1: Yes

Reviewer #2: Yes

Reviewer #3: Yes

5. Is the manuscript presented in an intelligible fashion and written in standard English?

Reviewer #1: Yes

Reviewer #2: Yes

Reviewer #3: Yes

6. Review Comments to the Author

Reviewer #1: My second comment was not addressed, so I repeat it.

2. How the authors would explain the antimicrobial activity of SeNP-modified fabric

materials if nanoparticles are strongly immobilized on their surfaces, thus hindering the

interactions with bacterial cells? Some positive and negative controls should be

performed with SeNPs and untreated fabric materials.

Reviewer #2: (No Response)

Reviewer #3: The authors have adequately addressed reviewer's comments and therefore it is suitable for publication.

7. PLOS authors have the option to publish the peer review history of their article (what does this mean?). If published, this will include your full peer review and any attached files.

Reviewer #1: No

Reviewer #2: No

Reviewer #3: No

---

## [Author Response · Author response to Decision Letter 1]

4 Nov 2024

Reviewer #1: My second comment was not addressed, so I repeat it.

2. How the authors would explain the antimicrobial activity of SeNP-modified fabric

materials if nanoparticles are strongly immobilized on their surfaces, thus hindering the

interactions with bacterial cells? Some positive and negative controls should be

performed with SeNPs and untreated fabric materials.

Response: Thank you for the comment! The article describes the microbiological study and the results obtained during the experiment. Even though selenium nanoparticles are included on the surface of the suture material, selenium still exhibits antimicrobial activity. We have added information on the probable mechanism of action of selenium nanoparticles. Detailed analysis of the mechanism of action of selenium nanoparticles on microorganisms is planned for future works. 

New changes are marked by red.

---

## [Decision Letter · Decision Letter 2]

7 Nov 2024

Synthesis and characterization of selenium nanoparticles stabilized with oxyethylated alkylphenol (neonol) for potential modification of fabric materials

PONE-D-24-17851R2

Dear Dr. Saleh Al-maaqar,

We’re pleased to inform you that your manuscript has been judged scientifically suitable for publication and will be formally accepted for publication once it meets all outstanding technical requirements.

Kind regards,

Nayan Ranjan Singha, Ph.D.

Academic Editor

PLOS ONE

Additional Editor Comments (optional):

Reviewers' comments:

Reviewer's Responses to Questions

**Comments to the Author**

1. If the authors have adequately addressed your comments raised in a previous round of review and you feel that this manuscript is now acceptable for publication, you may indicate that here to bypass the “Comments to the Author” section, enter your conflict of interest statement in the “Confidential to Editor” section, and submit your "Accept" recommendation.

Reviewer #1: All comments have been addressed

2. Is the manuscript technically sound, and do the data support the conclusions?

Reviewer #1: Yes

3. Has the statistical analysis been performed appropriately and rigorously? 

Reviewer #1: Yes

4. Have the authors made all data underlying the findings in their manuscript fully available?

Reviewer #1: Yes

5. Is the manuscript presented in an intelligible fashion and written in standard English?

Reviewer #1: Yes

6. Review Comments to the Author

Reviewer #1: (No Response)

7. PLOS authors have the option to publish the peer review history of their article (what does this mean?). If published, this will include your full peer review and any attached files.

Reviewer #1: No

---

## [Editor Report · Acceptance letter]

15 Nov 2024

PONE-D-24-17851R2 

PLOS ONE

Dear Dr. Al-maaqar, 

I'm pleased to inform you that your manuscript has been deemed suitable for publication in PLOS ONE. Congratulations! Your manuscript is now being handed over to our production team.

Kind regards, 

on behalf of

Dr. Nayan Ranjan Singha 

Academic Editor

PLOS ONE